# Evolution of Statistical Strength during the Contact of Amorphous Polymer Specimens below the Glass Transition Temperature: Influence of Chain Length

**DOI:** 10.3390/ma16020491

**Published:** 2023-01-04

**Authors:** Yuri M. Boiko

**Affiliations:** Laboratory of Physics of Strength, Ioffe Institute, 26 Politekhnicheskaya Str., 194021 St. Petersburg, Russia; yuri.boiko@mail.ioffe.ru

**Keywords:** amorphous polystyrene, contact, auto-adhesion strength, statistics, molecular weight

## Abstract

A comprehensive study of the statistical distribution of the auto-adhesion lap-shear strength (*σ*) of amorphous polymer–polymer interfaces using various types of statistical tests and models is a useful approach aimed at a better understanding of the mechanisms of the self-healing interface. In the present work, this approach has been applied, for the first time, to a temperature (*T*) range below the bulk glass transition temperature (*T*_g_^bulk^). The interest of this *T* range consists in a very limited or even frozen translational segmental motion giving little or no chance for adhesion to occur. To clarify this issue, the two identical samples of entangled amorphous polystyrene (PS) with a molecular weight (*M*) of 10^5^ g/mol or 10^6^ g/mol were kept in contact at *T* = *T*_g_^bulk^ − 33 °C for one day. The as-self-bonded PS–PS auto-adhesive joints (AJ) of PSs differing in *M* by an order of magnitude were fractured at ambient temperature, and their *σ* distributions were analyzed using the Weibull model, the quantile-quantile plots, the normality tests, and the Gaussian distribution. It has been shown that the Weibull model most correctly describes the *σ* statistical distributions of the two self-bonded PS–PS AJs with different *M* due to the joints’ brittleness. The values of the Weibull modulus (a statistical parameter) *m* = 2.40 and 1.89 calculated for PSs with *M* = 10^5^ and 10^6^ g/mol, respectively, were rather close, indicating that the chain length has a minor effect on the *σ* data scatter. The Gaussian distribution has been found to be less appropriate for this purpose, though all the normality tests performed have predicted the correctness of the normal distribution for these PS–PS interfaces.

## 1. Introduction

Generally, the arithmetic mean values of a certain mechanical property (e.g., strength *σ*_av_, Young’s modulus *E*, strain at break *ε*_b_) are estimated from a rather limited number (3–5) of measurements of identical samples used for materials’ characterization (see, e.g., Refs. [1,2,3,4]). However, a larger number of tests – numbering in the dozens, for instance–undoubtedly provides not only higher precision regarding the estimated characteristics, but also supports their reliability through additional parameters such as the standard deviation of the mean. In this case, moreover, an opportunity to investigate the statistical distribution of the chosen property appears. In turn, learning the most correct form of this distribution can give complementary useful information aimed at in-depth analysis of the deformation and fracture mechanisms in materials of various chemical origins. Indeed, on one hand, for the materials characterized by both extremely high (*σ* > 1 GPa [5,6,7], both organic and inorganic fibers) and extremely low strengths (*σ* < 1 MPa [8,9,10,11], weak polymer–polymer interfaces), i.e., for the materials drastically differing in the *σ* value, by 3 orders of magnitude, the *σ* statistical distributions have been found to follow the same distribution form: the Weibull distribution [12,13,14,15,16,17,18,19]. In this case, one observes a linear plot in the specific coordinates lnln [1/(1 − *P*_j_)] = f(ln*σ*), where *P*_j_ is the cumulative probability of failure. These distribution behaviors seem to be unsurprising since these two completely different materials types have one common feature, brittleness, and the Weibull model was initially proposed for such materials systems [12]. On the other hand, the materials with ‘intermediate strengths’, of the order of some dozens of MPa, which include many commercially-available polymers of general use such as polystyrene (PS), poly(2,6-dimethyl-1,4-phenylene oxide) (PPO), PS-PPO blends [20], and others, have demonstrated the statistical distribution behaviors of several mechanical properties (*σ*, *E*, and *ε*_b_), which are characteristic of the Gaussian distribution representing the bell curve. It is interesting to note that these statistical behaviors have been observed despite completely different deformation ability of these polymers (quasi-brittle PS and PS-PPO blends, and ductile PPO).

It is important to note that the validity of a given distribution type implies a certain mechanism of deformation or fracture. For instance, if the Weibull distribution for *σ* holds [12,13,14,15,16,17,18,19], it indicates that the fracture process is dominated by the presence of surface cracks and their propagation across the sample cross-section [6]. By contrast, if the Gaussian model is most correct, it suggests that the deformation or fracture process is controlled by many factors which are independent and equally weighted [21,22,23]. Therefore, by finding out which type of the statistical distribution is most appropriate, one may conclude which of the mechanisms dominates in the case considered.

Of particular interest for the investigation of mechanical properties is the statistical analysis of the strength development at the early stages of self-bonding at polymer–polymer interfaces, when two polymer pieces are brought into contact at a rather low temperature *T*, well below the bulk glass transition temperature (*T*_g_^bulk^), and kept at that *T* for a chosen period of time (*t*). During this period, the interface strength develops from “zero value” upon the contact (van der Waals interactions between the molecular groups located at the sample surface are too weak to give rise to practical adhesion [24]) to a certain value originating from the physical links formed by the interdiffusing chains ends (also mainly van der Waals interactions, but taking place between a larger number of links per contact area). Although this molecular mechanism is believed to be non-realistic at *T* < *T*_g_^bulk^, it can be activated in a *T* interval *T*_g_^interface^ < *T* < *T*_g_^bulk^ if *T*_g_^interface^ < *T*_g_^bulk^, as was suggested by Boiko and Prud’homme [24] and further developed by Boiko in [25]. In fact, the strength developed at these *T* conditions is rather low (<1 MPa [24]). Nevertheless, it is sufficient for the interface to bear a mechanical load. Consequently, this strength origination represents the early stages of the phenomenon of self-bonding at *T* < *T*_g_^bulk^.

Indeed, in the absence of the segmental interdiffusion across the interface, the interface resistance to a mechanical load is provided exclusively by wetting forces. In terms of the interface fracture energy (*G*), the value of *G* is equal to the thermodynamic work of adhesion (*W*_a_) required to reversibly separate the surfaces-in-contact; for the two fully identical samples, *W*_a_ = 2*γ* where *γ* is the free surface energy. For the majority of polymers, *W*_a_ is very low, *W*_a_ < 0.1 J/m^2^ [26]. For instance, since *γ* = 0.042 J/m^2^ for PS, one obtains *W*_a_ = 0.084 J/m^2^ for the two contacting PS samples. This *W*_a_ value is smaller by over an order of magnitude compared to the smallest value *G* = 2 J/m^2^ measured in the T-peel test geometry after self-bonding of a PS–PS interface at a rather low *T* = *T*_g_^bulk^ − 43 °C [27]. It indicates that this *G* value was mainly developed due to the contribution of the chain segments’ interdiffusion at this low *T*, which, at first glance, is rather unexpected. However, this behavior seems to be explainable in view of its occurrence above the local *T*_g_ of the interface layer (and, hence, of the surface layer prior to contact).

It should be noted that the impact of some molecular and structural factors such as the chain architecture, and the presence or the absence of crystallites on the statistical adhesion strength of weak polymer–polymer interfaces, has already been investigated [8,9,10,11]. However, little is known about the influence, if any, of such an important molecular factor as the chain length on the adhesion strength statistics. Moreover, the statistical lap-shear strength developed at the amorphous polymer–polymer interfaces at *T* < *T*_g_^bulk^ has not yet been analyzed on its validity to normal (or Gaussian) distribution [21,22,23]. Thus, in order to address these issues, the goal of this study is twofold: to find out (i) which of the types of the well-known statistical distributions is most correct to describe the distribution of the lap-shear strength (*σ*) developed during the contact of the two identical pieces of an amorphous polymer, and (ii) how the chain length affects this process at a rather low temperature, which was by 33 °C below *T*_g_^bulk^. For this purpose, two atactic (amorphous) PSs differing in chain length by an order of magnitude, with molecular weights (*M*) *M* = 10^5^ and 10^6^ g/mol, were used. The as-self-bonded PS–PS auto-adhesive joints were fractured at ambient temperature, and their *σ* distributions were analyzed using the Weibull model, the quantile-quantile plots, normality tests, and the Gaussian distribution.

## 2. Materials and Methods

### 2.1. Polymers and Samples

The polymers used in this study were two near-monodisperse PSs with a number-average molecular weight (*M*_n_) and a weight-average molecular weight (*M*_w_) of *M*_n_ = 97,000 g/mol and *M*_w_ = 102,500 g/mol, and *M*_n_ = 965,600 g/mol and *M*_w_ = 1,110,500 g/mol, abbreviated as PS1 and PS2, respectively, purchased from Polymer Source, Inc. (Dorval, QC, Canada). The *T*_g_^bulk^s of the two PSs were measured on a differential scanning calorimeter Q1000 (TA Instruments, New Castle, DE, USA) at a heating rate of 10 °C/min and estimated as the middle points of the corresponding heat capacity jumps at 105 °C (PS1) and 106 °C (PS2). The amorphous PS thick bulk films with a thickness of 100 μm were produced by compression molding of the PS powders between smooth surfaces of a silica glass at *T* = 165 °C (PS1) and *T* = 180 °C (PS2). The as-produced films were cut into rectangular strips of a width of 5 mm and a length of 30 mm.

### 2.2. Self-Bonding of PS–PS Interfaces

The PS strips were self-bonded (i.e., PS1 with PS1, and PS2 with PS2) at *T* = 72 °C (PS10^5^) and *T* = 73 °C (PS10^6^), i.e., at *T* = *T*_g_^bulk^ − 33 °C for the two polymers investigated, for 24 h. To form the PS–PS single lap-shear auto-adhesive joints (AJs) that are capable of bearing a mechanical load, 10 pairs of the identical amorphous PS strips were held in contact at an overlapped length of 5 mm and set in a fan-equipped oven. To facilitate good wetting, a small contact pressure of 0.2 MPa was applied to each of the contact areas individually by dead loads. This made it possible to avoid an undesirable non-uniformity of the contact pressure from joint to joint. To obtain reliable results, each self-bonding procedure was repeated. Thus, 20 identical joints were formed and tested for each of the PSs used.

### 2.3. Fracture Tests

The as-formed PS–PS AJs were fractured on an Instron tensile tester at ambient temperature at a crosshead speed of 5 mm/min. The distance between the tester clamps was 50 mm with the joint located in the middle. The auto-adhesion lap-shear strength (*σ*) was calculated as the AJ fracture load divided by the contact area of 25 mm^2^. The scheme of the procedures of the self-bonding interface and fracture is shown in Figure 1.

### 2.4. Statistical Analysis

#### 2.4.1. Weibull’s Statistics

The Weibull analysis of the lap-shear strength distribution was carried out using Equation (1) [12,13,14,15,16,17,18,19]:lnln [1/(1 − *P*_j_)] = −*m*·ln*σ*_0_ + *m*·ln*σ*,(1)
where *P*_j_ = (*j* − 0.5)/*n*, *n* is the auto-adhesive joint number, *m* is the Weibull modulus (or the shape parameter), and *σ*_0_ is the scale parameter. Equation (1) can be simplified to:*y* = *a* + *bx*,(2)
where *y* = lnln [1/(1 − *P*_j_)], *b* is the *m*, *x* is the ln*σ*, and *a* = −*m*·ln*σ*_0_ is the curve intersection with the *y* axis. By estimating *m* as the tangent to the curve lnln [1/(1 − *P*_j_)] = f(ln*σ*) using the standard procedure of the linear regression analysis, one can calculate *σ*_0_ as *σ*_0_ = exp(−*a*/*m*); *σ*_0_ is an equivalent of *σ*_av_.

#### 2.4.2. Normality Tests

To investigate the validity of the normal distribution, the *σ* data sets measured were analyzed–first, by constructing the normal probability (NP) or quantile-quantile (Q-Q) plots, and second, by computing them using several standard normality test procedures (Kolmogorov–Smirnov, Shapiro–Wilk, Lilliefors, Anderson–Darling, D’Agostino–K squared, and Chen–Shapiro tests) [22,23]. Thereafter, the histograms of the probability density function (PDF) vs. *σ* were constructed and analyzed to determine their correspondence to the Gaussian distribution.

## 3. Results and Discussion

In Figure 2a, the values of *σ* evolved at the PS1–PS1 and PS2–PS2 interfaces at *T* = *T*_g_^bulk^ − 33 °C for 24 h and measured thereafter at ambient temperature are plotted in ascending order. As is seen, the *σ* values for the PS1–PS1 joints are markedly higher as compared to those for the PS2–PS2 joints—approximately twice as high. The next step was to determine whether this *σ* difference with *M* agreed with Wool’s minor chain reptation model predicting *σ*~1/*M*^1/4^ [26] which is valid for self-bonding between polymers at *T* > *T*_g_^bulk^, when the long-range chain snake-like motion is feasible. According to this approach, one can write *σ* = *k*/*M*^1/4^ where *k* is a constant depending on the chain chemical structure. Hence, for the two PSs investigated, the ratio between *σ*(PS1) and *σ*(PS2), *σ*(PS1)/*σ*(PS2), has the form *σ*(PS1)/*σ*(PS2) = [*M*_n_(PS2)/*M*_n_(PS1)]^1/4^. To put it differently, the value of *σ*(PS1) reduced to that of *σ*(PS2) can be expressed as *σ*(PS1) = [*σ*(PS2)] × (965,600/97,000)^1/4^, i.e., it can easily be calculated as *σ*(PS1) = 1.776*σ*(PS2). Compare the calculated and measured values of *σ*(PS1) in Figure 2b. As is seen, the two *σ* data sets overlap. This observation supports the validity of the minor chain reptation model at *T* < *T*_g_^bulk^, though it was proposed initially for the temperature range *T* > *T*_g_^bulk^. This apparent contradiction can be omitted in view of the fact that the PS surface *T*_g_ is lowered with respect to the PS *T*_g_^bulk^ by roughly 50 °C [24], and by assuming that this effect persists at the early stages of self-bonding [25], the activation of the snake-like chain motion at a temperature which is not markedly lower than *T*_g_^bulk^, at *T* = *T*_g_^bulk^ − 33 °C, seems to be realistic if it occurs at *T* > *T*_g_^interface^.

The diffusion-controlled mechanism of the lap-shear strength evolution at *T* = *T*_g_^bulk^ − 33 °C can further be confirmed as follows. When the interface self-bonding process at this *T* was interrupted by removing the samples-in-contact from the oven after a short-term exposure for 1 min, thus minimizing or even excluding the interdiffusion contribution and only the wetting contribution to the lap-shear strength development remaining active, the upper sample was separated easily from the bottom sample just by careful handling. To put it differently, in this case, the adhesion force of physical attraction between the surfaces (*F*_a_) is smaller than the sample weight which is equal to the product of its volume (0.01 × 0.5 × 3 cm^3^) and density (~1 g/cm^3^), which gives *F*_a_ = 0.015 gf or 0.147 mN. Hence, for a contact area (*S*) of 25 mm^2^ used in the self-bonding experiments here, one obtains a higher limit of the adhesion strength *σ*_a_ = *F*_a_/*S* ≤ 5.9 × 10^−6^ MPa for the PS–PS interfaces investigated. This value of *σ*_a_ (corresponding to *G* = *W*_a_ = 0.084 J/m^2^ in terms of fracture energy) is markedly, by four order of magnitude, smaller than the smallest value of *σ* = 0.03 MPa measured in this work. Therefore, even this smallest *σ* value was developed due to the build-up of new van der Waals bonds between the molecular groups of the chain segments diffused from one PS sample and those of the counter PS sample they penetrate. Their concentration per unit of the PS–PS contact area has been estimated to be roughly 0.02 nm^−2^ at *T* = *T*_g_^bulk^ − 33 °C [28].

Let us turn to the statistical analysis of the measured lap-shear strength data sets. First, consider the applicability of the Weibull model. For this purpose, the data of Figure 2a were replotted as lnln [1/(1 − *P*_j_)] vs. ln*σ* in Figure 3 and investigated using a linear regression analysis.

As follows from Figure 3, the experimental data points for the two PSs investigated are fitted with rather high values of root mean square deviation *R*^2^ = 0.982 (*M* = 10^5^) and *R*^2^ = 0.987 (*M* = 10^6^). Hence, the results of this procedure are correct. The values of the Weibull modulus calculated from these plots are *m* = 1.89 and 2.40 for the PSs with *M* = 10^6^ and 10^5^, respectively. A higher *m* value for the PS1 with a lower molecular weight indicates that the data scatter for it is narrower. However, the difference between these two *m* values of ~20% seems to be rather small, especially with respect to the drastic difference in *M* and in the chain end concentrations (an order of magnitude). Therefore, although the chain ends represent structural defects, their increased concentration at the interface does not play a negative role in the *σ* data scatter.

Now, investigate the correspondence of the data of Figure 2a to the normal distribution. For this purpose, these data are presented as the quantile–quantile plots in Figure 4a,b for AJ PS1−PS1 and PS2−PS2, respectively. As is seen, the two data sets can be fitted satisfactorily with the linear curves, suggesting that they can be appropriate for representing the normal distributions.

To investigate this issue in more detail, these data sets were analyzed using several normality tests, and the results of this analysis are presented in Table 1 and Table 2.

As follows from the analysis performed, all the normality test types give the same result—“cannot reject normality”, assuming that the *σ* statistical distributions of the two PSs investigated are expected to follow the Gaussian distribution, i.e., to have the form of the bell curve. Therefore, the final conclusion concerning the validity of the Gaussian distribution can be received after the construction of the histograms PDF(*σ*) for the AJs of PS1 and PS2. The results of this procedure are shown in Figure 5, and it is seen that they do not correspond to the classical symmetrical bell curves with well-defined maxima.

Nevertheless, qualitatively, the histograms constructed for the two PSs investigated can be fitted at first approximation with a Gaussian function, albeit with various degrees of success. A better fitting result is observed for PS1 (see Figure 5a). This observation correlates with the smaller *p*-values for the PS1−PS1 AJ as compared to those for the PS2−PS2 AJ, in particular, for the Shapiro–Wilk and Anderson–Darling tests (see Table 1): 0.150 and 0.261, and 0.187 and 0.298 for PS1 and PS2, respectively. In fact, the *p*-values estimated mean there is a >15% chance of finding a result less close to expectation for the AJ PS1−PS1 while that for the AJ PS2−PS2 of >26% indicates a less reliable result for the latter (the Shapiro–Wilk test). By analogy, basing on the Anderson–Darling test results, the probabilities to find a less reliable result are >19% and >30% for PS1 and PS2, respectively. One may also notice the correlation for the sharpness and asymmetry of the peak (Kurtosis and Skewness in the D’Agostino–K squared test): the peak for PS2 is less symmetrical (larger *p*-value) and sharper (smaller *p*-value) with respect to that for PS2. Thus, one may conclude that both the Weibull analysis and the normality tests indicate that the more reliable results are obtained for the PS1−PS1 AJ.

## 4. Conclusions

For the first time, the comprehensive statistical analysis of the lap-shear strength *σ* evolved in the course of self-bonding between the two contacting PS samples realized via the chains interdiffusion across the PS–PS interface below *T*_g_^bulk^, at the rather low *T* = *T*_g_^bulk^ − 33 °C, has been performed. It has been shown that the *σ* distributions of the two PSs with drastically differing chain lengths, by an order of magnitude, can be correctly described using the Weibull standard distribution function. The values of the Weibull modulus *m* = 2.40 and 1.89 calculated for the two PSs with *M* = 10^5^ and 10^6^ g/mol, respectively, are fairly close despite the marked difference in *M*. However, a small 20 percent increase in the *m* value (i.e., a decrease in the data scatter) with a decrease in *M* indicates that the larger number of the interdiffusing chain ends in the polymer with a smaller *M* value makes the self-bonding process more uniform. The Gaussian distribution has been found to be less appropriate to correctly describe the *σ* distribution at the two PS–PS interfaces involved, despite the fact that all the normality tests performed have predicted the correctness of the normal distribution. A better suitability of the Weibull model for this purpose is because the PS–PS AJs were weak and quasi-brittle, and this model was first proposed namely for this class of materials. The results presented in this work stimulate further investigations in the field of the statistical distribution of the lap-shear strength developed after self-bonding at *T* < *T*_g_^bulk^ of the PS–PS interfaces over broader temperature intervals, down to *T* = *T*_g_^bulk^ − 80 °C, and of the interfaces of the polymers with other chain architectures using the combined approach involving various statistical tests and models.

## Figures and Tables

**Figure 1 materials-16-00491-f001:**
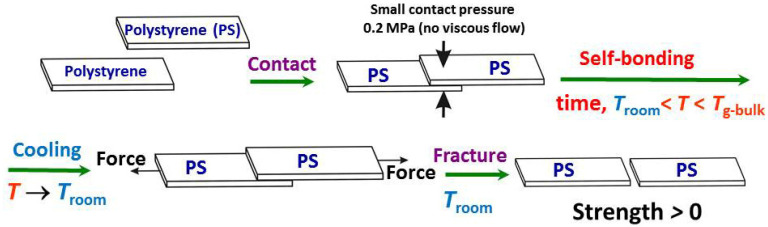
Scheme of the experimental procedures used in this work.

**Figure 2 materials-16-00491-f002:**
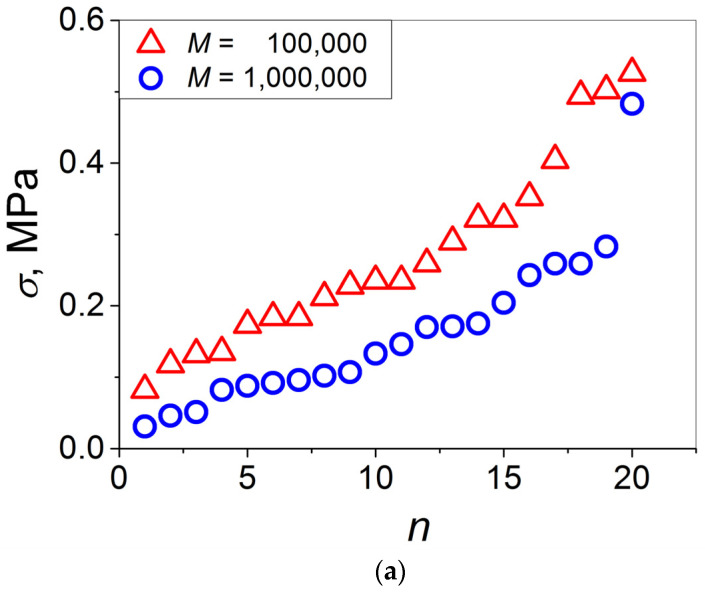
(**a**) Lap-shear strength *σ* ascending order as a function of joint number developed at symmetric PS–PS interfaces with (open triangles) *M* = 10^5^ and (open circles) *M* = 10^6^; (**b**) data of Figure 2a where the *σ* values for PS2 with *M* = 10^6^ are reduced to the *σ* values for PS1 with *M* = 10^5^ by the multiplication of the formers by (965,600/97,000)^1/4^ (see text).

**Figure 3 materials-16-00491-f003:**
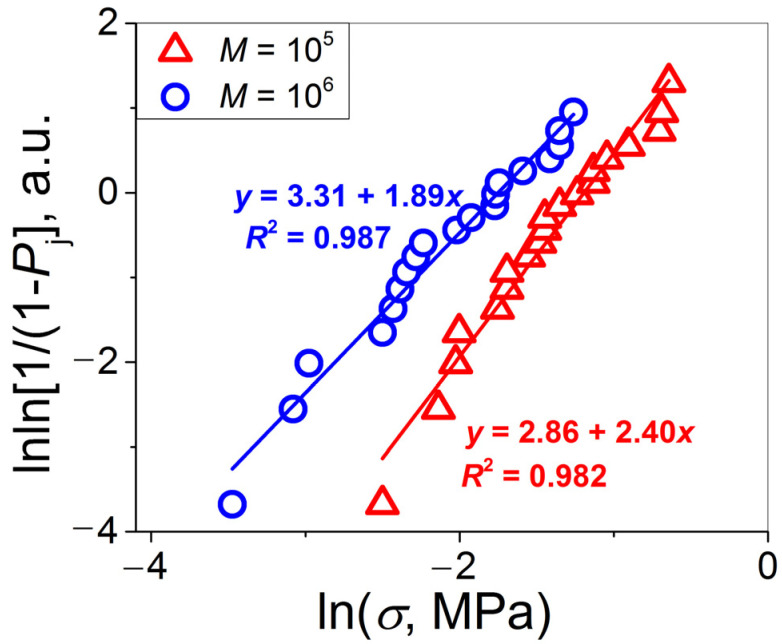
Weibull plots for the data presented in Figure 2a.

**Figure 4 materials-16-00491-f004:**
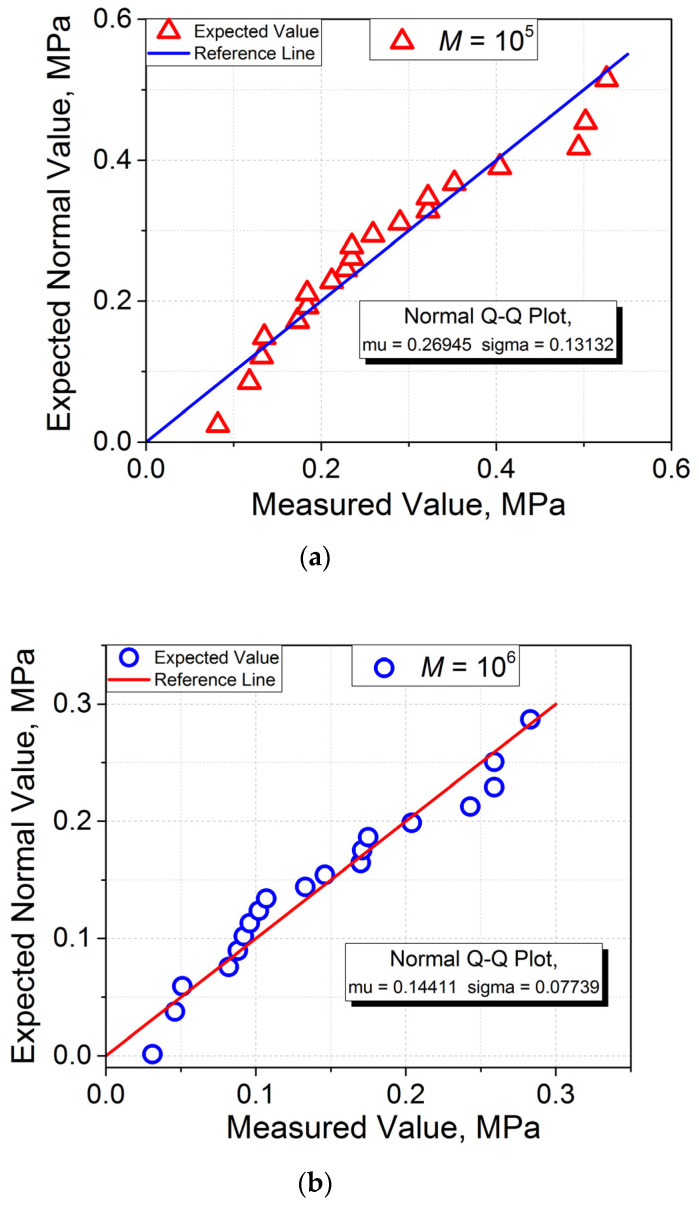
Quantile–quantile plots for (**a**) AJ PS1–PS1 and (**b**) AJ PS2–PS2.

**Figure 5 materials-16-00491-f005:**
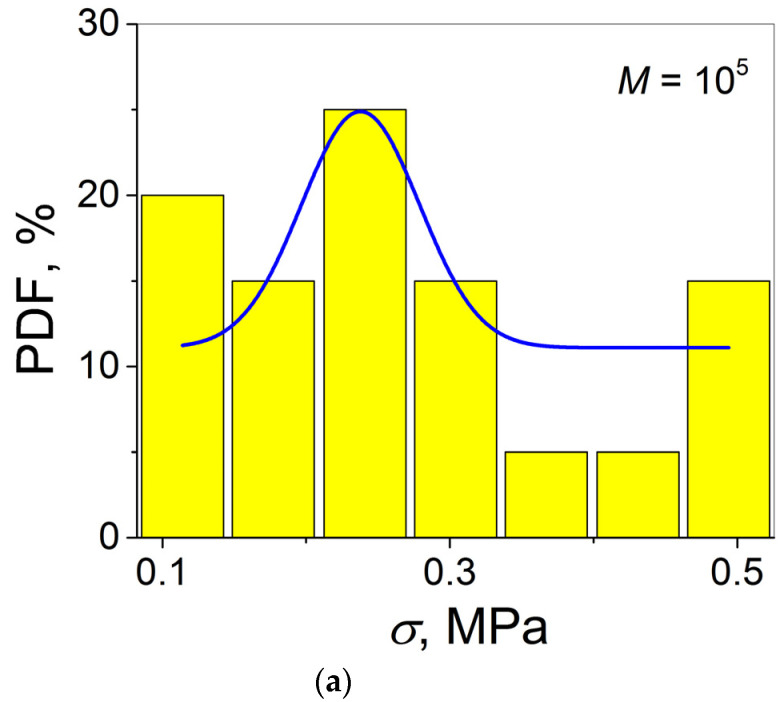
Histograms PDF vs. *σ* for AJs (**a**) PS1–PS1 and (**b**) PS2–PS2.

**Table 1 materials-16-00491-t001:** Statistical parameters of the lap-shear strength distribution estimated for two PS–PS interfaces in five normality tests.

Molecular Weight, g/mol	Test Type	Statistic	*p*-Value	Decision at Level 5% *
10^5^	Shapiro–Wilk	0.92935	0.15004	+
10^6^		0.93979	0.2614	+
10^5^	Lilliefors	0.15347	0.2	+
10^6^		0.15788	0.2	+
10^5^	Kolmogorov–Smirnov	0.15347	0.70769	+
10^6^		0.15788	0.70223	+
10^5^	Anderson–Darling	0.49761	0.18727	+
10^6^		0.41675	0.29844	+
	D’Agostino–K squared:			
10^5^	Omnibus	1.95367	0.3765	+
	Skewness	1.36904	0.17099	+
	Kurtosis	−0.28178	0.77811	+
10^6^	Omnibus	1.94532	0.37808	+
	Skewness	0.79625	0.42589	+
	Kurtosis	−1.14512	0.25216	+

* “+” in column 5 means “cannot reject normality”.

**Table 2 materials-16-00491-t002:** Statistical parameters of the lap-shear strength distribution for two PS–PS interfaces estimated from the Chen–Shapiro normality test.

Molecular Weight, g/mol	Statistic	10% Critical Value	5% Critical Value	Decision at Level 5% *
10^5^	−0.00868	0.01371	0.05109	+
10^6^	−0.03627	0.01427	0.05232	+

* “+” in column 5 means “cannot reject normality”.

## Data Availability

Not applicable.

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
