# Peer review of "Evolution of Statistical Strength during the Contact of Amorphous Polymer Specimens below the Glass Transition Temperature: Influence of Chain Length"

_materials, 2023, doi:10.3390/ma16020491_

Round 1

Reviewer 1 Report

In this line of research, the author investigated the evolution of statistical strength during the contact of amorphous polymer specimens below the glass transition temperature, which has rarely been explored previously. Two samples of amorphous PS with different molecular weights were used to obtain the auto-adhesion lap-shear strength of the as-self-bonded PS-PS auto-adhesive joints formed below Tg. It was concluded that the Weibull’s model is most correct for describing the σ statistical distributions. Interestingly, the σ values differ greatly with different M due to the joints’ brittleness, while the Weibull’s moduli are fairly close despite the marked difference in M. The research is enlightening, and the results are convincing. Therefore, I would like to recommend its publication in Materials after the following minor concerns are addressed.

1.     A specific temperature T=Tg-33 oC was chosen. Why? Would varying the temperature below Tg affect the conclusion?

2.     It would be helpful if the author could give a perspective on how other types of polymers would behave below Tg.

Author Response

Reviewer 1

In this line of research, the author investigated the evolution of statistical strength during the contact of amorphous polymer specimens below the glass transition temperature, which has rarely been explored previously. Two samples of amorphous PS with different molecular weights were used to obtain the auto-adhesion lap-shear strength of the as-self-bonded PS-PS auto-adhesive joints formed below Tg. It was concluded that the Weibull’s model is most correct for describing the σ statistical distributions. Interestingly, the σ values differ greatly with different M due to the joints’ brittleness, while the Weibull’s moduli are fairly close despite the marked difference in M. The research is enlightening, and the results are convincing. Therefore, I would like to recommend its publication in Materials after the following minor concerns are addressed.

  1. A specific temperature T=Tg-33 oC was chosen. Why? Would varying the temperature below Tg affect the conclusion?

- This temperature was chosen for the following reasons. First, because it is located actually below Tg-bulk (e.g., T = Tg-10 oC would be not convincing as T < Tg-bulk since it can be treated as the Tg surroundings). Second, the lap-shear strength developed at this temperature is not very low, as it is after self-bonding at Tg-50 oC. However, this is an interesting issue to be further investigated. This work is planned in the future. The corresponding text is added in the Conclusion.

  1. It would be helpful if the author could give a perspective on how other types of polymers would behave below Tg.

- Other polymers (PMMA, PPO) behave below Tg-bulk in a similar way from the point of view of Weibull’s distribution (see refs 9 and 10). However, the normal distributions of their lap-shear strength have not been investigated yet. This work is planned in the future. The corresponding text is added in the Conclusion: The results received in the present work stimulate further investigations in the field of the statistical distribution of the lap-shear strength developed at T < Tg-bulk after self-bonding of the PS-PS interfaces over broader temperature interval, down to an extremely low T = Tg-bulk – 80 oC, and of the interfaces of the polymers with other chain architectures using the combined approach involving various statistical tests and models.

Reviewer 2 Report

(1) The significance of this research should be further explored. If we want to bond two samples together, a relatively high temperature should be applied, or selecting the materials with low glass transition temperature or melting point. As shown in Figure 2, the strength of joints is really low, which makes it useless. Therefore, the results of this work are not attracting.

(2) In the equation 1, n is the joint number, how can you get this parameter? In the section 2.2, only the overlapped length can be obtained.

(3) Surface evenness play a critical role in bonding, authors should characterize the surface evenness of samples and control it in the same level.

Author Response

Reviewer 2

(1) The significance of this research should be further explored. If we want to bond two samples together, a relatively high temperature should be applied, or selecting the materials with low glass transition temperature or melting point. As shown in Figure 2, the strength of joints is really low, which makes it useless. Therefore, the results of this work are not attracting.

- Actually, the joints’ strength developed below Tg is low. However, the fact of its occurrence is itself very interesting because it seems to be unrealistic a priori, under conditions (T < Tg) when the long-range chain segments’ motions are frozen, which do not allow them to cross the contact zone and to build-up new physical links between the contacting samples. These results are very interesting and important in the scientific, fundamental sense, in particular, for the investigation of the surface and interface phenomena.   

(2) In the equation 1, n is the joint number, how can you get this parameter? In the section 2.2, only the overlapped length can be obtained.

- Joint number is obtained after sorting the lap-shear strength values in the ascending order. For 20 identical joints tested, the joint number 1 has the lowest strength while the joint number 20 has the highest strength. 

(3) Surface evenness play a critical role in bonding, authors should characterize the surface evenness of samples and control it in the same level.

- Of course, surface roughness plays an important role in adhesion. However, for the purpose of the present work aiming at the lap-shear strength distribution analysis, this factor is not of critical importance. Most importantly is to use the samples of one and the same surface roughness, and namely such identical samples produced by compression molding between smooth silica glass plates were used in this study. 

Reviewer 3 Report

This is review for the ‘Evolution of Statistical Strength during the Contact of Amorphous Polymer Specimens below the Glass Transition Temperature: Influence of Chain Length’ by Yuri M. Boiko. This study has been focused on figuring out which of the types of well-known statistical distributions is most correct to describe the distribution of stress developed during the contact of the two identical pieces of amorphous polymers. Moreover, the author has been proposed how the chain length influence the self-bonding process of polymers. Overall, the work appears precisely performed and interpreted.  To improve this work, however, reviewer would like to recommend major revision.  Before accepting for publication, the following comments need to be addressed.

1 The abstract is slightly long, however, no critical points about this study was found. The abstract needs to include discussion about major objective; the summary of the results; and major findings. What is the relevance and novelty of this study?

2 In the introduction part, references are partly missing and the fundamental knowledge about characterizing interfacial strength and shear strength development between two different molecule groups is missing. Reviewer would like to recommend providing technical details and fundamental concepts of interfacial strength and shear strength development between two different molecule groups. A discussion of this point in the light of previous literature should be included as an outlook: ‘Biomacromolecules 2017, 18 (9), 2876’, ‘Nature nanotechnology 2018, 13 (11), 1057’, ‘Composites Part B, 2019,160, 535’, ‘Composites Part A 2016, 82, 53’.

3. This manuscript is lack of materials characterization. As fundamental standard data, please provide the following data:

(i) Raman data to show chemical structures and component interactions. OR;

(ii) TEM or SEM images to show interfacial organization/structures. OR;

(iii) XRD results to similarly show interfacial organization/structures.

4. Surface interactions and adhesion forces data about other different types of polymers need to be mentioned for identifying typical polymers’ forces at the interface in terms of relevance. 

Author Response

Reviewer 3 

1 The abstract is slightly long, however, no critical points about this study was found. The abstract needs to include discussion about major objective; the summary of the results; and major findings. What is the relevance and novelty of this study?

- The abstract is modified in accordance with your suggestions and comments. The novelty of the study consists in the performance, for the first time, of a comprehensive statistical analysis of the lap-shear strength developed below Tg-bulk using several statistical approaches, and in the investigation of the influence of the chain length on the statistical behavior.    

2 In the introduction part, references are partly missing and the fundamental knowledge about characterizing interfacial strength and shear strength development between two different molecule groups is missing. Reviewer would like to recommend providing technical details and fundamental concepts of interfacial strength and shear strength development between two different molecule groups.

-  In the absence of the segmental interdiffusion, the interface fracture energy (G) which is provided exclusively by wetting forces (the work of adhesion Wa) is very low for the majority of polymers: Wa < 0.1 J/m^2 [ref 27]. This Wa value is smaller by an order of magnitude as compared to the smallest value G = 2 J/m2 measured after self-bonding of a PS-PS interface at a rather low T = Tg-bulk – 43 oC [new ref 28]. It means that this G value was developed due to the contribution of the chain segments interdiffusion even at such low T. However, this behavior seems to be reasonable in view of its occurrence above the local Tg of the interface layer (and, hence, of the surface layer prior to contact). This text is added in the Introduction, pages 2-3.

- Also see below response to Comment 4. 

  1. This manuscript is lack of materials characterization. As fundamental standard data, please provide the following data:

(i) Raman data to show chemical structures and component interactions. OR;

(ii) TEM or SEM images to show interfacial organization/structures. OR;

 (iii) XRD results to similarly show interfacial organization/structures.

- The chemical structure of polystyrene is well known, it represents a carbon-carbon chain with a phenyl side group. Besides, the segmental motions take place in a nm-thick interface layer, Raman resolution is not sufficient. Amorphous atactic PSs were used in this study, no structure organization, X-Ray diffraction cannot be helpful.

- The purpose of the work was to investigate the statistical behavior of the mechanical properties, of the interface strength, and to analyze it, for the first time, using several classical approaches. Most importantly for this comparative study (to find out in which way the interface strength is distributed statistically inside of one population) was to use one and the same contacting samples of each PS in order to define correctly the most appropriate type of the statistical distribution.

  1. Surface interactions and adhesion forces data about other different types of polymers need to be mentioned for identifying typical polymers’ forces at the interface in terms of relevance.

- From the point of view of the work of adhesion Wa (the energy required to reversibly separate the contacting surfaces), the wetting contribution for the majority of polymers is very low, < 0.1 J/m^2. For comparison, the smallest values of the fracture energy reported for PS-PS interfaces are 1-2 J/m^2 [new ref 28]. Further, if any similar or dissimilar polymers such as PS, PMMA, PET and many others are brought into contact at small pressures like used in the present work, no ‘practical’ adhesion will occur between them, and they will be separated without applying any external mechanical force, just by handling carefully the upper contacting sample. To put it differently, the ‘practical’, i.e. realistic, adhesion force is equal to “zero” because the adhesion forces are not capable to resist to the separation force which is equal to the sample) weight of 0.015 g (volume  0.5 x 3 x 0.01 cm^3, density 1 g/cm^3). In other words, the adhesion strength at the contact area of 25 mm2 is smaller than 6 x 10^-6 MPa – this is the result of the attraction between the molecular groups located on the two contacted PS surfaces. In this work, the smallest measured strength value is 0.03 MPa which is higher than the (overestimated) adhesion strength = 6 x 10^-6 MPa by four orders of magnitude. This is the result of the intermolecular interaction between the interdiffused chain segments of one sample and the polymer matrix chain segments of the counter-sample they penetrate. The concentration, per unit of the contact area, of the fractured van-der-Waals bonds which were built-up as a result of this process at T = Tg – 33 oC has been estimated to be roughly 0.02 nm^-2 [new ref 29]. The corresponding text is added in the Results and Discussion, page 5.

Round 2

Reviewer 3 Report

The authors have stated and revised most of the previously mentioned questions. However, the description of the interfacial strength and shear strength development between two different molecule groups is too vague. It is hard to estimate the chain segments interdiffusion by wetting force since the interfacial strength can be affected by several factors. Reviewer keeps try to ask for scientific details and fundamental concepts of interfacial strength and shear strength development between two different molecule groups. The main reason for the increasing overall strength of polymer materials is the increase of the interfacial strength at the interface. Thus, the quantitative analysis approaches of interfacial strength need to be considered and interpreted.

Author Response

Responses to Comments of Reviewer 3

The authors have stated and revised most of the previously mentioned questions. However, the description of the interfacial strength and shear strength development between two different molecule groups is too vague. It is hard to estimate the chain segments interdiffusion by wetting force since the interfacial strength can be affected by several factors. Reviewer keeps try to ask for scientific details and fundamental concepts of interfacial strength and shear strength development between two different molecule groups. The main reason for the increasing overall strength of polymer materials is the increase of the interfacial strength at the interface. Thus, the quantitative analysis approaches of interfacial strength need to be considered and interpreted.

  • There is no connection between the wetting forces (forces of physical attraction between the two contacting PS surfaces) and the lap-shear strength developed due to interdiffusion. The first, wetting, contribution follows exclusively from the van-der-Waals attraction forces between the molecular groups located on the two contacting surfaces. In contact are one and the same polymers the chains of which have one and the same chemical structure (PS) and, therefore, one and the same, not different, molecular groups. The wetting forces between the two PS surfaces, as expressed by the thermodynamic work of adhesion Wa, are equal to 2g where g is the free surface energy: Wa = 2g. For PS, g = 0.042 J/m^2, which makes Wa = 0.084 J/m^2. The measured fracture energy G = 2 J/m^2 for the PS-PS interface after the contact at T = Tg – 43 oC includes two contributions: 0.084 J/m^2 from wetting and (2 - 0.084) J/m^2 = 1.916 J/m^2 from interdiffusion. To put it differently, the wetting contribution can be neglected. The corresponding text is modified in the Introduction, page 2, bottom, page 3, top, and in the Results and Discussion, page 5.   
  • Moreover, when in contact are the polymers with different chemical structures, and, hence, different molecule groups, e.g., PS and PMMA, Wa will also be very small, < 0.1 J/m^2. In this case, Wa = g(PS) + g(PMMA) - g(PS/PMMA) which is even smaller than Wa (PS).
  • All these adhesion mechanisms are considered in detail in reference 27 [Wool, R.P. Polymer Interfaces: Structure and Strength; Hanser Press: New York, NY, USA, 1995].

Round 3

Reviewer 3 Report

 Accept in present form